# Development of Dispersion Process to Improve Quality of Hyaluronic Acid Filler Crosslinked with 1,4-Butanediol Diglycidyl Ether

**DOI:** 10.3390/polym16233323

**Published:** 2024-11-27

**Authors:** Sunglim Choi, Jin Cheol Cho, Seunghwa Lee, Seong Jin Lee

**Affiliations:** CHA Meditech Co., Ltd., 119 Techno 2-ro (#206, Migeun Techno World, Yongsan-Dong), Yuseong-gu, Daejeon 34116, Republic of Korea; cho928@chamc.co.kr (J.C.C.); lshlsh0106@chamc.co.kr (S.L.); sjlee@chamc.co.kr (S.J.L.)

**Keywords:** hyaluronic acid, BDDE crosslinker, dermal filler, dispersion process

## Abstract

This study proposes a new and simple process that improves the quality of a hyaluronic acid (HA) filler crosslinked with 1,4-butanediol diglycidyl ether (BDDE) using solution dispersion at a low temperature. This process involves the solvent being dispersed among the solute naturally after the mixing process. The process used in this study involved two reactions. First, the solution was dispersed among HA molecules (Mw = ~0.7 MDa) creating a well-homogenized mixture. Second, the decomposition and synthesis of HA occurred naturally in an aqueous alkaline solution (>pH 11), the weight average molar mass (Mw) was adjusted (Mw = ~143,000), and the crosslinking surface area was expanded, allowing for a high degree of crosslinking. Therefore, the viscoelasticity and cohesion of the filler increased with the new method compared to the previous process both at the lab scale (previous process:new process, viscosity (cP) = 24M:43M, storage modulus (Pa) = 306:538, loss modulus (Pa) = 33:61, and tack (N) = 0.24:0.43) and at the factory scale (previous process:new process, complex viscosity (cP) = 19M:26M, storage modulus (Pa) = 229:314, loss modulus (Pa) = 71:107, and tack (N) = 0.35:0.43).

## 1. Introduction

Dermal fillers are generally used to remove wrinkles, restore volume, and provide moisture. They are injected directly into skin layers; therefore, they must be made of materials that are either similar to skin tissue or do not cause adverse reactions [1,2,3,4,5,6,7,8,9]. In the early stages of dermal filler development, materials such as collagen [2], poly methyl methacrylate (PMMA) [3], and dextran [4] were used to manufacture the fillers. One advantage of using these materials to produce fillers is the fact that they are semi-permanent; however, they are difficult to remove from the different layers of the skin. If these materials cause adverse reactions, such as allergies, surgical treatment may be required to remove them from the skin [5], which is disadvantageous in terms of both cost and health.

At present, most dermal fillers are made using hyaluronic acid (HA). This choice of material is based on the fact that HA is decomposed by enzymes such as hyaluronidase, and the substances that are produced in the process do not cause harm to the body. However, the advantages and disadvantages of using HA are identical; HA is decomposed in the body, and, as a result, the filler material is not maintained in the skin layer [6,7,8,9,10]. To address the above disadvantage, various crosslinkers have been reacted with HA, with this process increasing the duration of the filler remaining in the skin layer [10,11,12,13].

Among them, 1,4-butanediol diglycidyl ether (BDDE) is commonly used as a crosslinker because the crosslinking method and required conditions are simple; the ring-opening reaction of the epoxy group in BDDE occurs at a relatively low temperature. In addition, the reaction requires only a basic (pH > 11) solution [8,13,14].

However, the manufacturing of HA fillers crosslinked with BDDE (HA-BDDE crosslinked fillers) faces three difficulties.

First, when poly HA is dissolved in an aqueous solution at a high concentration, common methods such as stirring, are limited to produce a homogeneous solution [15,16,17,18]. Therefore, the compound must be mixed manually using a Teflon bar; however, this method is prone to human error. At present, the use of planetary mixers means that this problem has, for the most part, been solved [19]. However, it is difficult to produce a homogeneous solution.

Second, the above problem means that a relatively large amount of unreacted BDDE with HA remains in the mixture or is hydrolyzed. This, then, results in the filler having low yield, elasticity, and viscosity [20,21].

Third, the remaining BDDE and oxidized BDDE in the filler are toxic and can cause allergic reactions. Additionally, pendant BDDE, in which one side of the epoxy group of BDDE reacts with the primary alcohol group within a single HA unit, may also lead to other similar side effects [21,22]. For this reason, ensuring the correct crosslinking ratio between HA and BDDE is especially important. Furthermore, as the crosslinking ratio increases, the elasticity and viscosity of the filler increase, which improves the quality of the filler [9].

For the above reasons, by mixing HA and a NaOH solution, an attempt was made to produce a more homogeneous mixture compared to those made using other processes. [23]. HA is a hydrophilic compound, and the aqueous solution can disperse among the HA chains. Therefore, the ‘dispersion’ process was hereby introduced into filler production; this provided enough time for the NaOH solution to disperse among the poly HA chains at a suitable temperature. This process is expected to produce a homogeneous material.

Chemically, HA chains are connected and cleaved simultaneously in pH > 11 due to hydrolysis and dehydration [24]. If the reaction rate of hydrolysis and dehydration of HA can be controlled, HA with appropriately low number-average molar mass (Mn) and weight-average molar mass (Mw) can be produced. This process will enable the production of poly HA chains of an appropriate length, ensuring a sufficient surface area for the reaction with BDDE. However, if the HA chains become too short, the elasticity, viscosity, and cohesion of the fillers will decrease [25]. Therefore, by controlling the temperature and duration of the ‘dispersion’ process, the optimal conditions for this process can be determined.

Due to the efficiency of the dispersion process, the time of the dispersion process is limited to one day, and the temperature of this process is lower than room temperature (rt).

In this study, the optimal time and temperature of the dispersion process at a lab scale were determined and introduced into the production process of the filler. Thereafter, the elasticity, viscosity, and cohesion of the filler materials were measured to determine the improvement in filler quality [26,27]. In addition, the degree of modification (MoD) was measured to determine the improvement in the crosslinking ratio between HA and BDDE [28].

## 2. Methods of Synthesis and Analysis

### 2.1. Synthesis of the HA-BDDE-Crosslinked Filler

#### 2.1.1. Batch A

First, 15 g of HA ([η] = 1.4, η = intrinsic viscosity, Hyundai Bioland Co., Ltd., Cheongju, Republic of Korea, Mw = ~0.7 MDa) was placed into a planetary centrifugal mixer case, and 0.25 N of NaOH solution was added to produce 17 wt% of HA solution based on the molecular weight of sodium hyaluronate. Next, the mixture was blended using a planetary centrifugal mixer (ARE-500, THINKY, Chiyoda, Japan) for 10 min at 1000 rpm. This process was repeated twice. Thereafter, the mixture was blended for 10 min at 1000 rpm and deformed for 5 min at 2000 rpm. At this point, the mixer case was maintained at a low temperature using a cooling adapter. The reference mixture was synthesized following the previous process [22], as shown in Figure 1.

The experimental mixture was subjected to dispersion processes with different temperatures and durations.

When BDDE (Sigma-Aldrich Co., Ltd., St. Louis, MO, USA) was added to the mixture, the concentration was 3.5 mol%. Next, the mixture was blended using the planetary centrifugal mixer for 5 min at 1000 rpm. This process was repeated twice. Thereafter, the mixture was deformed for 2 min at 2000 rpm. At this point, the mixer case was maintained at a low temperature using a cooling adapter.

The mixture was poured into the metal frame (20 × 20 × 20 mm by a cell, 120 × 120 × 20 mm by a frame), and the frame was placed in an oven at 25 °C for 24 h. The mixture was transformed into an elastic gel form. The gels (20 × 20 × 20 mm by a gel) were placed in dialysis bags. Afterward, the dialysis bags were placed in 0.9 (*w*/*v*)% NaCl solution, and the solution was slowly stirred for 5 days. The solution was replaced twice a day.

Lastly, the hydrogels (50 × 50 × 45 mm by a gel) were crushed to 300 μm using the mesh sieve (DH.Si8038, DAIHAN Scientific, Incheon, Republic of Korea), and the crushed hydrogels were sterilized using an autoclave (HS-3041VD, HANSHIN MEDICAL Co., Ltd., Incheon, Republic of Korea) at 120 °C for 20 min.

#### 2.1.2. Batches B and C

In this step, 160 g of HA ([η] = 1.4, η = intrinsic viscosity, Hyundai Bioland Co., Ltd., Republic of Korea, Mw = ~0.7 MDa) (Batch B) or 320 g (Batch C) was placed into the planetary mixer case, and 0.25 N of NaOH solution was added to produce 17 wt% of HA solution based on the molecular weight of sodium hyaluronate. Next, the mixture was blended twice using a planetary centrifugal mixer (OST-CMT1811, OSTAR TECH Co., Ltd., Pyeongtaek, Republic of Korea), and it was ultimately deformed for 10 min at 600 rpm. At this point, the mixer was maintained at −6 °C.

The reference mixture was synthesized following the previous process, as shown in Figure 1.

The experimental mixtures were subjected to the dispersion process at 18 °C for 24 h. When BDDE was added to the mixture, the concentration was 3.5 mol%. To ensure that the BDDE was mixed thoroughly, the mixing process was carried out in the same manner as above. The mixture was poured into the metal frame, and the frame was placed in the oven at 25 °C (Batch B) or 30 °C (Batch C) for 24 h. The mixture was transformed into an elastic gel form. The gels (20 × 20 × 20 mm by a gel) were placed in 0.9 (*w*/*v*)% NaCl solution (Batch B) or 1× phosphate-buffered saline (PBS) pH 7.4 solution (Batch C) using a swelling device (PID-101, SEJIN T&E Co., Ltd., Jeonju, Republic of Korea), and the solution was stirred slowly for 5 days. The solution was replaced twice a day.

Lastly, the hydrogels (50 × 50 × 45 mm by a gel) were crushed to 300 μm using a plunger mill (OST-PM60-A01, OSTAR TECH Co., Ltd., Republic of Korea), and the crushed hydrogels were sterilized using an autoclave (BP-AC680G, Bio Plan, Daejeon, Republic of Korea) at 120 °C for 20 min.

### 2.2. Decomposition of HA-BDDE Crosslinked Hydrogel

First, 1 g of the crushed hydrogel was place in a vial (>4 mL) and 1 mg of chondroitinase ABC from Proteus vulgaris (2 unit/mg, Sigma-Aldrich Co., Ltd., USA) was dissolved in 1× PBS solution pH 7.4 (20× PBS solution (LPS Solution, Daejeon, Republic of Korea)): Distilled water = 19:1) 10 mL. Thereafter, a solution of 1.2 mL was added to the vial containing the hydrogel. After the mixture was stirred for 30 min, the vial was placed in a shaking incubator set to 157 rpm for 5 days. Next, the vial was placed in an ultralow-temperature freezer (MDF-192, SANYO Electronic Co., Ltd., Osaka, Japan) for 1 h. Lastly, the mixture was dried using a freeze-dryer (FD8508, ilShinBioBase Co., Ltd., Dongducheon, Republic of Korea) for 3 days [29].

### 2.3. Measurement of Viscoelasticity and Cohesion

Using a rheometer (Phisica MCR301, Anton Paar, Ostfildern, Germany), complex viscosity, storage modulus (G′), loss modulus (G″), and tack were measured.

Next, 1 mL of filler material was placed on the ground plate of the rheometer at 25 °C, and, using a rheometer software (Rheoplus V. 3.62, Anton Paar, Germany), the measuring plate (Φ = 25 mm) was pulled down until the gap with the ground plate was 1 mm (sample size: Φ25 × 1 mm). Thereafter, complex viscosity, G′, and G′′ were measured. Lastly, tack was measured.

### 2.4. Measurement of Gel Filtration Chromatography (GFC) and Nuclear Magnetic Resonance (NMR)

GFC (e2695, Waters, Pleasanton, CA, USA) commissioned the Korea Research Institute of Chemical Technology (KRICT) to conduct the measurements. To produce the GFC sample, 1 g of the crushed hydrogel was poured into a vial and placed in an ultralow-temperature freezer for 1 h. Thereafter, the mixture was dried using a freeze-dryer for 3 days. This powdered material was submitted to KRICT for GFC analysis.

NMR (AVANCE III 600, Burker, Billerica, MA, USA) was either conducted upon request by Chungnam National University or performed directly. To produce the NMR sample, 20 mg of the decomposed compound was dissolved in 1 mL of D_2_O (Cambridge Isotope Laboratories, Tewksbury, MA, USA) and poured into an NMR tube (508HP, Norell, Morganton, NC, USA) [29].

## 3. Results and Discussion

In previous studies, various methods were employed to enhance the viscoelasticity and cohesion of fillers. These included using HA with a relatively high Mw [30], incorporating compounds such as mannitol [31], and employing alternative crosslinkers like poly (ethylene glycol) diglycidyl ether (PEGDE) [32,33], and divinyl sulfone (DVS) [26,34].

However, this experiment was conducted to enhance the viscoelasticity and cohesion of the filler through simple process modifications, without altering the composition of specific substances, as mentioned above. Additionally, efforts were made to improve these properties beyond the results achieved in previous studies [35] conducted at this facility.

### 3.1. Experimental Results of Products Manufactured in the Lab

#### 3.1.1. Viscoelasticity and Cohesion

Complex viscosity represents the overall viscosity and elasticity of a material. G′ reflects its elasticity, whereas G″ indicates its viscosity.

In Figure 2a–c and Appendix A, complex viscosity, G′, and G″ show a tendency to increase over time, except at 4 °C. The graph of complex viscosity shows that the value for the previous experiment was roughly 2.4 McP, and the values for the experiment including the dispersion process were about 4.8 McP at 10 °C and 4.3 McP at 18 °C for 24 h—roughly twice the previous values.

The G′ and G″ of the experiment involving the dispersion process were also roughly twice those of the previous experiment. When comparing all values, the elasticity and viscosity of the filler material increased over time. Additionally, the cohesion of the filler materials was known through the tack and the complex modulus (G*). In Figure 2d,e and Appendix A, tack and G* can also be seen increasing over time. In the experiment including the dispersion process, the values were 0.41~0.43 for tack and 500~600 Pa for G*. These values were also roughly twice those of the previous experiment.

As shown in Figure 3a–c, the elasticity and viscosity increased up to 10 °C, after which the values slightly decreased until 18 °C. However, in Figure 3e, the tack values can be seen increasing with the temperature. There was no significant difference in elasticity and viscosity between 10 °C and 18 °C; however, the cohesion value changed, as increasing the tack value was very difficult.

When summarizing the above results, the time and temperature of the dispersion process were determined to be 24 h at 18 °C.

#### 3.1.2. Molar Mass

As mentioned above, relatively lowering the Mn and Mw of polymers increases their surface area. In Figure 4 and Table 1, the Mn and Mw of HA subjected to the dispersion process at 18 °C for 24 h are reduced to 1/6 compared to those of unprocessed HA.

It was assumed that the change in the molecular weight of HA had affected the crosslinking ratio between HA and BDDE [36]. This is because the lower the Mw, the larger the reaction surface area of the polymer. Through the changes in physical properties and Mw, it was believed that the process hereby described yielded a higher crosslinking ratio compared to the previous process.

### 3.2. Experimental Results of Products Manufactured in the Factory (Batch B)

#### 3.2.1. The Apparent State

Based on the above results, the dispersion process at 18 °C for 24 h was introduced at the production scale (Batch B). When simply blending HA in the 0.25 N NaOH solution using a planetary mixer, the mixture was cloudy, and there were also areas where HA was clumped together in powder form.

After the dispersion process, the mixture became clear; there were no places where HA accumulated. Therefore, HA that had undergone the dispersion process seemed to react better with BDDE than HA that had undergone the previous process (Figure 5).

It was assumed that the change in the polydispersity of HA affected the crosslinking ratio between HA and BDDE. However, there was no difference in their polydispersity; therefore, increasing the polydispersity of HA does not affect the molar mass of HA during dispersion. During the dispersion process, the reactive surface is expected to increase compared to untreated HA.

#### 3.2.2. Viscoelasticity and Cohesion

As shown in Figure 6a–c and Appendix A, the complex viscosity and G′ of products manufactured in the dispersion process at 18 °C for 24 h were 1.2 times higher than those of products with no dispersion; G″ was 1.1 times higher. As shown in Figure 6d,e and Appendix A, compared to the cohesion of the products, the tack and G* of products manufactured during the dispersion process at 18 °C for 24 h were 1.2 times higher than those of products with no dispersion. It can be concluded that the viscoelasticity and cohesion of products manufactured using the dispersion process were better than those of products with no dispersion.

As mentioned above, increasing the elasticity and viscosity indicated an improvement in the quality of the filler. However, if the values of the elasticity and viscosity are too high, the injection pressure of the filler will increase. It is therefore crucial to manufacture products that account for these factors. When measuring the injection pressure with a 27 G needle, it was found that the values were all similar and less than 20 N. In other words, as a result of the dispersion process, the quality of the filler increased.

#### 3.2.3. Degree of Modification (MoD)

The stability of the filler material is as important as its viscoelasticity and cohesion. The stability can be determined based on the crosslinking ratio, because the higher the crosslinking ratio, the more rigid the structure of the filler material. Therefore, the filler’s decomposition rate was slow. These results mean that the filler may remain in the body slightly longer. The crosslinking ratio was analyzed based on the degree of modification (MoD) and the ratio between the number of hydrogens in C7 and C6, 6′. This number can be checked in 1H NMR. The equation is as follows [29,37]:MoD (%) = (I^δ1.65~1.70^/4)/(I^δ2.00~2.10^/3)∙100(1)

The MoD (%) of the filler materials was limited to 3.5 because the concentration of the added BDDE was 3.5 mol%.

As shown in Figure 7 and Appendix A, the MoD of the manufactured products through the dispersion process was 3.43, and the product produced with no dispersion process was 3.16. The reason for these results is the fact that the crosslinking ratio was higher than in the previous product. These results therefore show that the stability of the fillers increased.

By combining the above results with this finding, the material produced through the dispersion process in this experiment showed improved quality as a filler.

### 3.3. Experimental Results of Products Manufactured in the Lab (Batch C)

#### 3.3.1. Viscoelasticity and Cohesion

Based on the above results, the dispersion process was introduced to the production process (batch C). In Figure 8a–c and Appendix A, complex viscosity and G′ increased by 1.4 times and G″ increased by 1.7 times when comparing the products manufactured through dispersion and the non-dispersion products. In Figure 8d,e and Appendix A, G* increased by 1.4 times and tack increased by 1.3 times in comparison.

Despite the changes in the crosslinking temperature of the BDDE and the purified solution during the purification process, the increases in the overall viscoelasticity and cohesion were greater than those observed in the previous process.

#### 3.3.2. Degree of Modification (MoD)

As shown in Figure 9 and Appendix A, the MoD of products manufactured using the new process increased more than in those manufactured using the old process. That is, the quality of the material produced through new process was better than that of the material made through the old process.

## 4. Conclusions

In the present study, a new and simple process was developed and implemented in a manufacturing process, even reaching the production stage. Using the dispersion process, three main achievements were accomplished:

First, the process reduced the burden placed on workers in relation to production and minimized human errors. As a result, workers no longer needed to mix the materials directly, to eliminate heterogeneous parts during the mixing process of HA and the base solution. It also helped to prevent human errors that could occur as a result of said mixing.

Second, physical properties such as the viscoelasticity and cohesion of the filler improved compared to existing products by introducing our simple process. As mentioned in the results above, the dispersion process was confirmed to increase the viscoelasticity and cohesiveness of the filler materials, and this improvement was maintained under all production conditions.

Lastly, the crosslinking ratio of the product increased. As the reaction surface area of HA increased through the dispersion process, it became easier for collision reactions to take place, which was confirmed by MoD (%). This finding demonstrated that the overall quality-related data improved, including enhancements in viscoelasticity and cohesion.

From a product-regulatory perspective, there was a limitation in not being able to change the product’s HA concentration in NaOH solution and the amount of crosslinking agent, so a clearer interpretation of the new process was lacking. However, the process of homogeneously dissolving a high concentration of HA could improve the crosslinking efficiency and physical properties of the final product.

Going forward, a method or process to improve the quality of the final products will be developed, and its applicability to other areas will be explored.

## 5. Patent

The patent application number is 10-2024-0120713 (Republic of Korea).

## Figures and Tables

**Figure 1 polymers-16-03323-f001:**
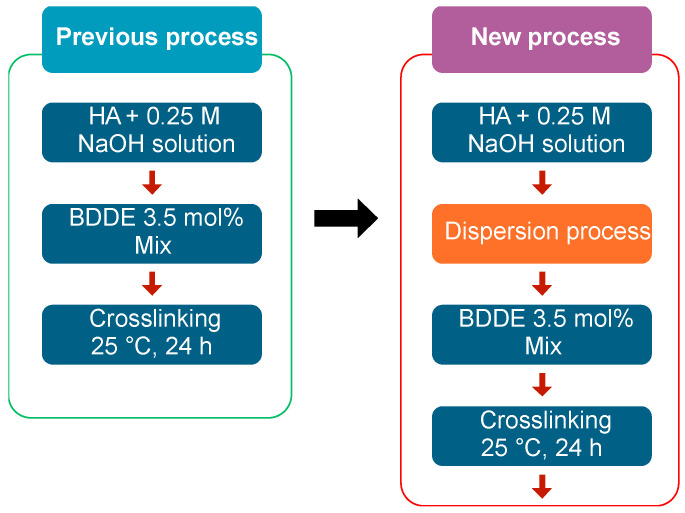
Chart of the difference between the previous process and the new process (dispersion process at a low temperature).

**Figure 2 polymers-16-03323-f002:**
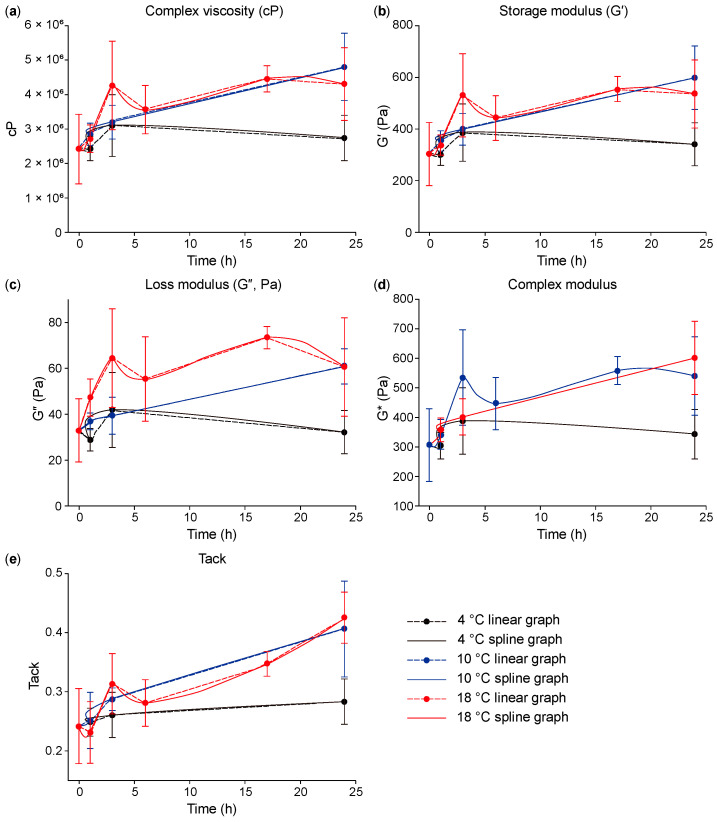
(**a**) Complex viscosity (cP), (**b**) storage modulus (G′), (**c**) loss modulus (G″), (**d**) complex modulus (G*), and (**e**) tack over dispersion time at each dispersion temperature (4 °C, 10 °C, and 18 °C, *n* = 3).

**Figure 3 polymers-16-03323-f003:**
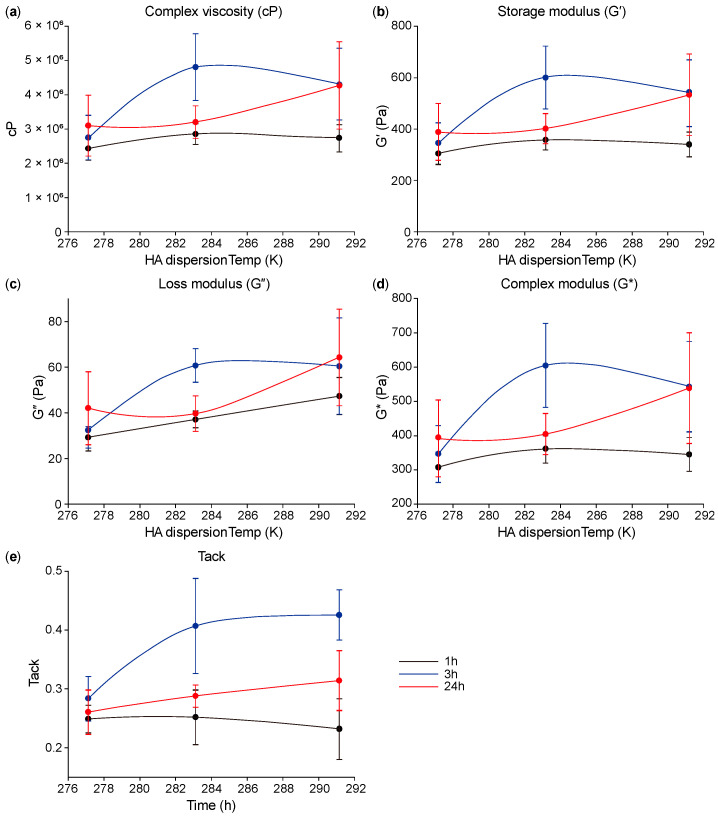
(**a**) Complex viscosity (cP), (**b**) storage modulus (G′), (**c**) loss modulus (G″), (**d**) complex modulus (G*), and (**e**) tack over dispersion temperature at each dispersion time (1 h, 3 h, and 24 h, *n* = 3).

**Figure 4 polymers-16-03323-f004:**
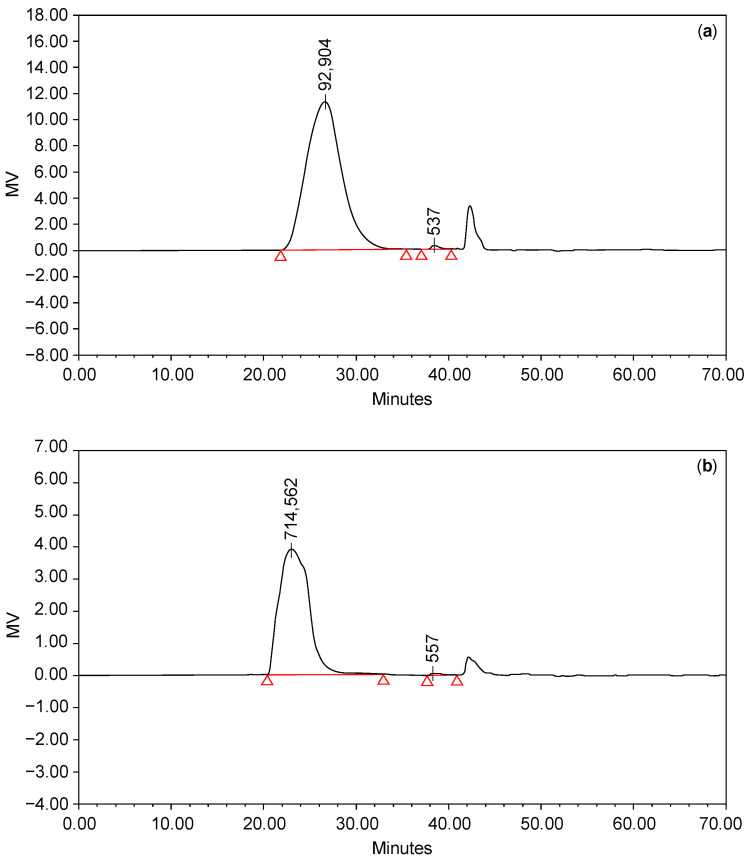
(**a**) GFC graph of 17 wt% HA in 0.25 N NaOH solution including the dispersion process (18 °C; 24 h); and (**b**) GFC graph of 17 wt% HA aqueous solution.

**Figure 5 polymers-16-03323-f005:**
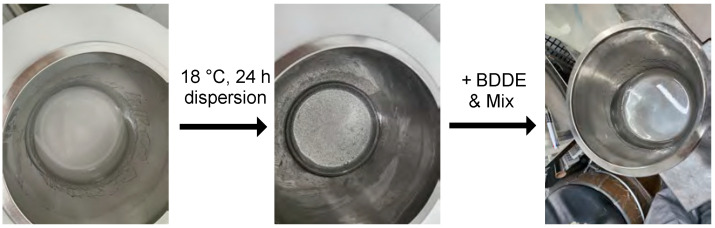
Visible transparency changes in HA in 0.25 N NaOH solution after the dispersion process (18 °C; 24 h) and then mixed with BDDE using a revolving–rotating mixer.

**Figure 6 polymers-16-03323-f006:**
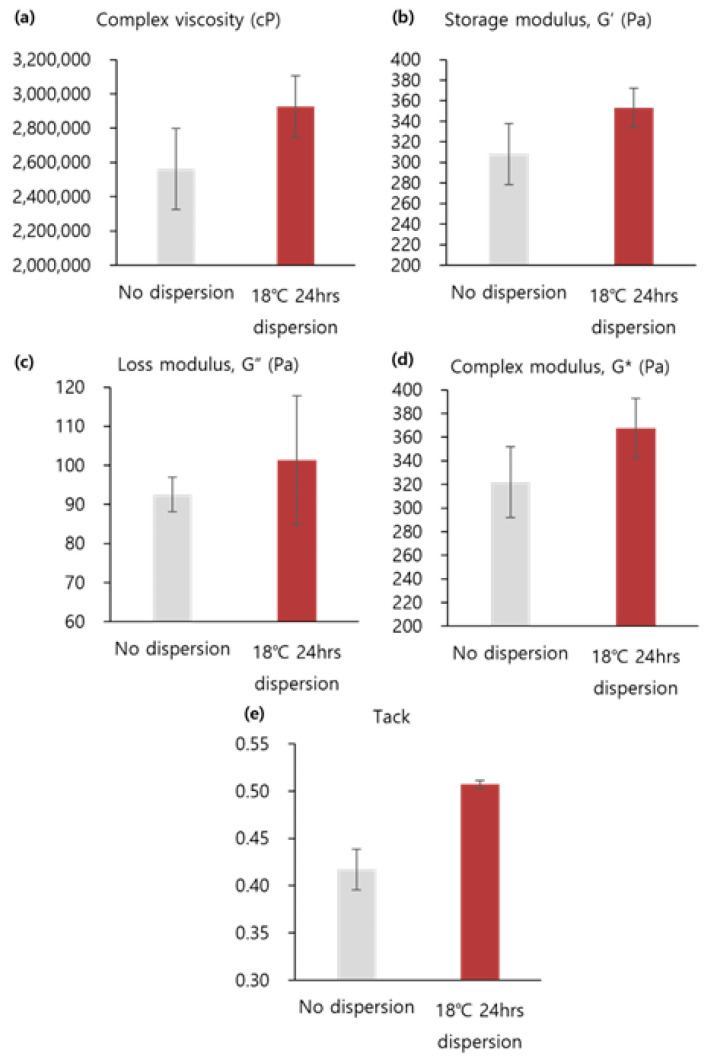
Comparison between old and new process at the 1/2 factory scale: (**a**) complex viscosity (cP), (**b**) storage modulus (G′), (**c**) loss modulus (G″), (**d**) complex modulus (G*), and (**e**) tack (*n* = 3).

**Figure 7 polymers-16-03323-f007:**
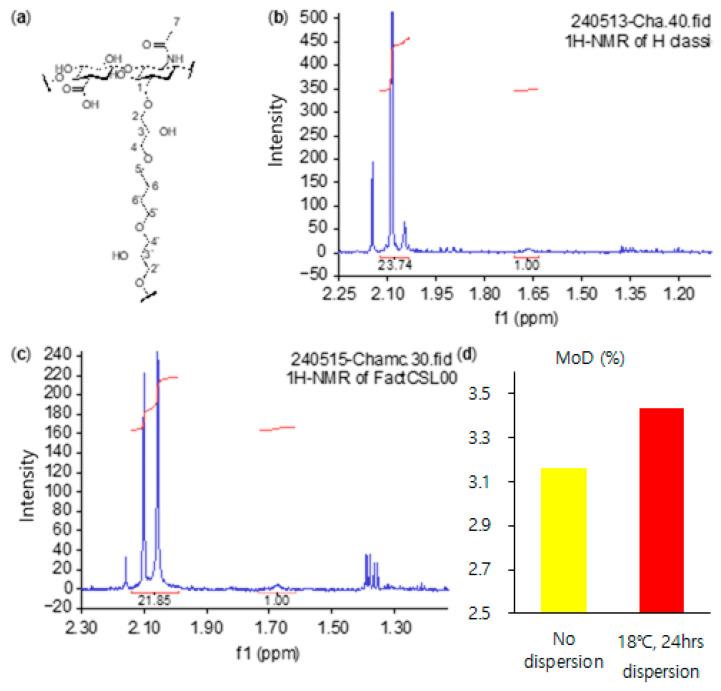
(**a**) The molecular structure of the BDDE-crosslinked HA dimer; (**b**) 1H NMR of the product being produced (no dispersion); (**c**) 1H NMR of the product made through a new process (18 °C and 24 h dispersion); and (**d**) graph detailing the MoD (%) of (**b**,**c**).

**Figure 8 polymers-16-03323-f008:**
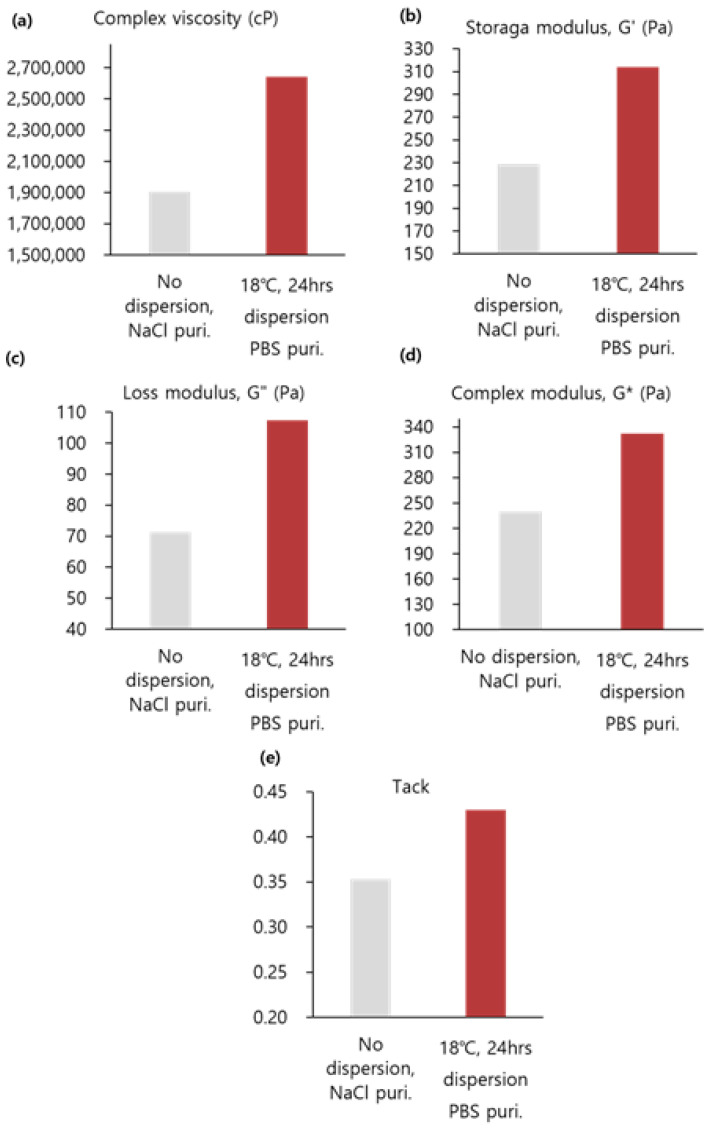
Comparison between the old and new process at the factory scale: (**a**) complex viscosity (cP), (**b**) storage modulus (G′), (**c**) loss modulus (G″), (**d**) complex modulus (G*), and (**e**) tack.

**Figure 9 polymers-16-03323-f009:**
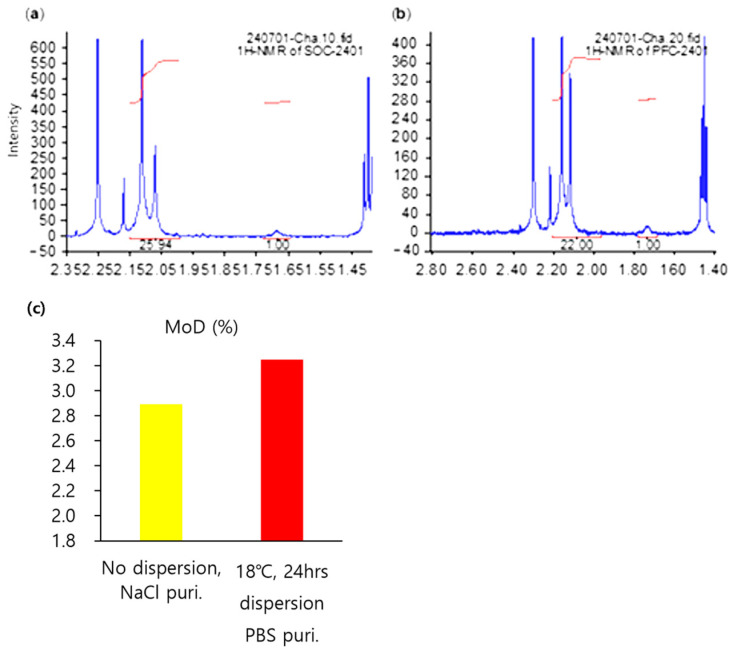
(**a**) Graph showing the 1H NMR of products made (with no dispersion) at the factory scale; (**b**) 1H NMR of the products made through the new process (18 °C and 24 h dispersion) at the factory scale; and (**c**) graph of the MoD (%) of (**a**,**b**).

**Table 1 polymers-16-03323-t001:** Number-average molecular weight (Mn), weight-average molecular weight (Mw), and polydispersity under (a) and (b) in Figure 4.

Process	Mn (Da)	Mw (Da)	Polydispersity
18 °C and 24 h dispersion	60,620	143,034	2.36
No dispersion	336,990	768,281	2.28

## Data Availability

The original contributions presented in the study are included in the article/Appendix A, further inquiries can be directed to the corresponding author.

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
