# Peer review of "Development of Dispersion Process to Improve Quality of Hyaluronic Acid Filler Crosslinked with 1,4-Butanediol Diglycidyl Ether"

_polymers, 2024, doi:10.3390/polym16233323_

Round 1

Reviewer 1 Report

Comments and Suggestions for Authors

Hyaluronic acid (HA), the main component of the extracellular matrix of joints, skin and cornea, is a linear polysaccharide with repeating units of D-glucuronic acid and N-acetyl-D-glucosamine linked alternately by β-1,3 and β-1,4 glycosidic bonds. HA is an attractive biopolymer due to its biocompatible, biodegradable, non-adhesive, non-immunogenic, hydrophilic and viscoelastic properties. HA has excellent qualities in skin regeneration, providing moisturizing and rejuvenating properties and is successfully used as a dermal filler. The importance and usefulness of practical application of hyaluronic acid in various forms, as well as the production of new medicinal forms with improved viscoelastic properties on its basis are beyond doubt.

Since HA is destroyed and synthesized in the body fairly quickly, modified versions of HA are used for fillers, in which various cross-linking agents react with HA, which increases the duration of the filler in the skin layer. In the peer-reviewed work, the authors described an improved process for obtaining HA-BDDE crosslinked filler by introducing an additional stage into the technological process. Thus, the HA production process was optimized to obtain a high-quality and effective product.

A distinctive feature of HA is the fact that in nature this biopolymer is synthesized with different molecular weights, and its functions are largely determined by its Mw.

The above suggests that:

1.      The initial material characterized by the phrase HA 1,4(Hyundai Bioland.Co., Ltd.) should be described in great detail in terms of origin and molecular weight. Although the molecular weight information was found in the abstract: (Mw=0.7~1.6MDa), it should definitely be in the Materials section of the main text.

2.      In the Materials section, the selected BDDE cross-linker is not characterized; there is not even a decipher of this designation.

3.      The molecular weight of hyaluronic polymer chains in HA-BDDE crosslinked filler was reduced by 6 times with the introduction of an additional process stage. How did the medical and biological properties of the modified hyaluronic acid change? For use in medicine, this is the question of primary importance. The study of physical (viscoelastic) properties should be relegated to the background.

4.      Review information on the medical and biological properties and their dependence on the molecular weight of hyaluronic polymer chains in HA-BDDE crosslinked filler should be provided in the text of the article, and the list of literary sources should be increased.

5.      Some designations were not deciphered, for example, PBC (line 115).

Author Response

Comment 1: The initial material characterized by the phrase HA 1,4(Hyundai Bioland.Co., Ltd.) should be described in great detail in terms of origin and molecular weight. Although the molecular weight information was found in the abstract: (Mw=0.7~1.6MDa), it should definitely be in the Materials section of the main text.

Response 1: Based on the above feedback, I revised the sentence on line 85, 112 and added additional content.

Comment 2 :In the Materials section, the selected BDDE cross-linker is not characterized; there is not even a decipher of this designation.

Response 2: Based on the above feedback, I revised the sentence on line 37 and added additional content.

Comment 3: The molecular weight of hyaluronic polymer chains in HA-BDDE crosslinked filler was reduced by 6 times with the introduction of an additional process stage. How did the medical and biological properties of the modified hyaluronic acid change? For use in medicine, this is the question of primary importance. The study of physical (viscoelastic) properties should be relegated to the background.

Comment 4: Review information on the medical and biological properties and their dependence on the molecular weight of hyaluronic polymer chains in HA-BDDE crosslinked filler should be provided in the text of the article, and the list of literary sources should be increased.

Response 3-4: 

As you mentioned above, the molecular weight of HA is very important and related to physical properties such as viscoelasticity and cohesion, even excluding the reaction by the cross-linking agent BDDE.

For HA of the same molecular weight, dissolution becomes more difficult at higher concentrations of HA (17 w/v% in this study) compared to lower concentrations due to its water-absorbing properties. In addition, at low concentration BDDE (3.5 mol v/v%, in this study) was prepared.

In this study, at high concentration of HA and low concentration BDDE, the cross-linking efficacy (MoD) was obtained the higher value than the previous method.

It would be expected for the more chance to reaction between the reduced low molecular chain by NaOH solution and two epoxides of BDDE.

So, additional method was introduced by ‘dispersion process’ in terms of more homogenous solution and low molecular weights by time and a certain temperature compared to the previous manufacturing method.

It would be expected the long duration in the body according to the improved physical properties than the previous process. In this study, the medical and biological properties was not studied unfortunately.  Additional biological and medical research will be performed.

Comment 5: Some designations were not deciphered, for example, PBC (line 115)

Response 5: Based on the above feedback, I revised the sentence on line 124-125 and added additional content.

Reviewer 2 Report

Comments and Suggestions for Authors

1. Please change and concise the title relying with the main finding and provide the full name of abbreviation.

2. Please clarify the meaning of "solution dispersion" in abstract.

3. The full name of chemicals should be informed before its abbreviation; numerical mean of finding parameters should be included in abstract.

4. The language check and improvement have to be conducted through all contents with third party, especially the grammatical concern.

5. Full name of chemicals should be come first before the abbreviation in the content.

6. For Introduction, please more literature review on your developed method both from research articles and patents applying to the other polymers of fillers. And how about their property improvement.

7. The detail of all chemicals and instruments should be revealed for their lot No., brand, company, city, state, country. For all chemicals they should be all included in Materials part as  section 2.1.

8. For all indicating of % please let us know its unit; % w/w , w/v???

9. Please include the sample size of experiments; with statistical analysis in Method and Result with the discussion of the main obtained data.

10. The evaluations related to the application as cosmetic filler should be also addressed in this manuscript for understanding its efficiency.

11. The final product comparison or obtained at each stage of productions should be shown with their photographs that clearly with high resolution or in suitable container for observation.

12. The discussion should be included the comparison with the related research works or with correlated supporting references.

13. For Fig. 6,8 please revise the format of graph as internation mean for presentation and statistical analysis.

14. Please consider for format style of Fig. 7d, 9c.

15. Please avoid using the word "we, our" in the content.

16. The claimmation about the limitation of this research work should be included in Conclusion. And please conclude into the single paragraph with the significant finding and suggestion.

Comments on the Quality of English Language

Language check and improvement have to be conducted through all contents with third party, especially the grammatical concern.

Author Response

Comment 1. Please change and concise the title relying with the main finding and provide the full name of abbreviation.

Response 1: Regarding the title, the focus of this paper was the increase in viscoelasticity and tack values observed by introducing a dispersion process between the mixing steps in the making HA-BDDE crosslinked filler process. The simplicity of this method was particularly notable, and the title emphasized this aspect. In other words, the title was crafted to reflect the primary development. Additionally, all abbreviations have been corrected accordingly.

Comment 2. Please clarify the meaning of "solution dispersion" in abstract.

Response 2: Based on the above feedback, I revised the sentence on line 10~11 and added additional content.

Comment 3. The full name of chemicals should be informed before its abbreviation; numerical mean of finding parameters should be included in abstract.

Response 3: We have provided the full names for all abbreviations and revised sections related to numerical values.

Comment 4. The language check and improvement have to be conducted through all contents with third party, especially the grammatical concern.

Response 4: The language issues of this paper were checked and improved through MDPI Author Services.

Comment 5. Full name of chemicals should be come first before the abbreviation in the content.

Response 5: This issue was checked and corrected accordingly.

Comment 6. For Introduction, please more literature review on your developed method both from research articles and patents applying to the other polymers of fillers. And how about their property improvement.

Response 6: We have reviewed the above issues, consulted additional papers as references, and included them in the reference section.

Comment 7. The detail of all chemicals and instruments should be revealed for their lot No., brand, company, city, state, country. For all chemicals they should be all included in Materials part as  section 2.1.

Response 7: This issue was checked and corrected accordingly.

Comment 8. For all indicating of % please let us know its unit; % w/w , w/v???

Response 8: This issue was checked and corrected accordingly. (0.9(w/v)% NaCl solution)

Comment 9. Please include the sample size of experiments; with statistical analysis in Method and Result with the discussion of the main obtained data.

Response 9: This issue was checked and corrected accordingly. (Part 2.3 and 2.4) The number of repetitions was added for each experiment.

Comment 10. The evaluations related to the application as cosmetic filler should be also addressed in this manuscript for understanding its efficiency.

Response 10: This information is found in lines 76–81 of the introduction and in references 26 and 27 of the paper.

Comment 11. The final product comparison or obtained at each stage of productions should be shown with their photographs that clearly with high resolution or in suitable container for observation.

Response 11: These issues of paper were checked and improved through MDPI Author Services.

Comment 12. The discussion should be included the comparison with the related research works or with correlated supporting references.

Response 12: In our paper, the focus is primarily on comparing our previous research methods, so we conducted comparisons with our past studies. Similarly, for the products, comparisons were made with those previously produced.

Comment 13. For Fig. 6,8 please revise the format of graph as internation mean for presentation and statistical analysis.

Comment 14. Please consider for format style of Fig. 7d, 9c.

Comment 15. Please avoid using the word "we, our" in the content.

Response 13-15: These issues of paper were checked and improved through MDPI Author Services and We discussed it among corrected ourselves accordingly . 

Comment 16. The claimmation about the limitation of this research work should be included in Conclusion. And please conclude into the single paragraph with the significant finding and suggestion.

Comment 16: From a product regulatory affair, there was a limitation in not being able to change the product's HA concentration in NaoH sol. and the amount of cross-linking agent, so a clearer interpretation of the new process would have been lacking. However, the process of homogeneously dissolving the high-concentration of HA could improve cross-linking efficiency and physical properties.

Round 2

Reviewer 1 Report

Comments and Suggestions for Authors

Manuscript may be accepted in present form.

Author Response

Dear reviewer,

We sincerely appreciate your time and effort in reviewing our manuscript titled "The Development of a New ‘Dispersion Process’ to Improve the Quality of Hyaluronic acid (HA) filler crosslinked with 1,4-butanediol diglycidyl ether (BDDE)”

We are delighted to hear your positive feedback and that the manuscript may be accepted in its present form. Your thoughtful comments and suggestions during the review process have been invaluable in refining our work, and we are grateful for your support and guidance.

Please do not hesitate to let us know if there are any additional steps required on our part to facilitate the final stages of the publication process.

Thank you once again for your valuable input and for helping to enhance the quality of our work.

Best regards,

Sunglim Choi, PhD

Manager

MD R&D, CHAMEDITECH. Co.

Fax 82-42-936-7096 Tel 82-10-9489-4489  e-mail:slchoi85@chamc.co.kr

Reviewer 2 Report

Comments and Suggestions for Authors

1. Please avoid to use "we, our, us" in Abstract, conclusion and other parts of content.

2. Full name of HA-BDDE should be addressed before abrreviation in title and abstract and Introduction.  

3. Title should be concassed as "Development of New Dispersion Process to Improve Quality of full name before HA-BDDE Crosslinked Fillers".

4. The numerical mean of main finding should be included in Abstract.

5.  HA 1.4, what is it?

6. Detail of brand, company,.. of planetary mixer planetary mixer planetary mixer and instrument of " steam sterilization" and " crushing" and others should be informed.

7. please consider for punctuation such as  2-3days. and so on in the content.

8.  (508HP, Norell, USA), which is the state?

9. There is still no have the sample size of examine in Method.

10. The discussion with comparison with related article, patent or research works is needed that might be other crosslinkers or chemicals and process with supporting references in all subsections of 3 as usual international style of research articles. Unless it is impossible to consider for publish.

11. Order of presenting equation should be ordered such as .....eq 1.

12.  Please add the unit of y-axis of Fig. 7b-c. and decimal of Fig7c should be only 1 decimal.

13. Please check and correct the use of , at the end of title or journal name in all ref lists for consistency and follow the journal format. doi?

14. Supplementary Materials:  ???

Author Response

  1. Please avoid to use "we, our, us" in Abstract, conclusion and other parts of content.

Response 1: This feedback has been reviewed, and some sentences addressing this point have been revised.

  1. Full name of HA-BDDE should be addressed before abrreviation in title and abstract and Introduction.  
  2. Title should be concassed as "Development of New Dispersion Process to Improve Quality of full name beforeHA-BDDE Crosslinked Fillers".

Response 2,3: This feedback has been reviewed, and the title addressing this point has been revised.

  1. The numerical mean of main finding should be included in Abstract.

Response 4: This feedback has been reviewed, and The numerical mean of main finding has been included in Abstract. (line 19-22)

  1. HA 1.4, what is it?

Response 5: The number following HA indicates its intrinsic viscosity. For instance, HA 1.4 refers to a high molecular weight HA polymer with an average intrinsic viscosity of 1.4 m³/kg. The intrinsic viscosity of a polymer serves as a key indicator for determining its weight-average molecular weight. A brief explanation regarding intrinsic viscosity has been included in the main text. This specific HA was used in all processes. (line 91, 122)

  1. Detail of brand, company of planetary mixer planetary mixer planetary mixer and instrument of " steam sterilization" and " crushing" and others should be informed.

Response 6: Detail of brand, company of planetary mixer was already commented (line 94-95, 126) the others have been reviced. (line 115-118, line 140-141)

  1. please consider for punctuation such as 2-3days. and so on in the content.

Response 7: This feedback has been reviewed, and some sentences addressing this point have been revised.

  1. (508HP, Norell, USA), which is the state?

Response 8: This feedback has been reviewed, and the state has been added. (line 173)

  1. There is still no have the sample size of examine in Method.

Response 9: This feedback has been reviewed, and sample size of the hydrogels and rheometer have been added. (line 110-112, line 115, line 135, line 139, line 158-160)

  1. The discussion with comparison with related article, patent or research works is needed that might be other crosslinkers or chemicals and process with supporting references in all subsections of 3 as usual international style of research articles. Unless it is impossible to consider for publish.

Response 10: This feedback has been reviewed, and the two paragraphs and some references have been added. (line 176-185)

  1. Order of presenting equation should be ordered such as .....eq 1.

Response 11: This feedback has been reviewed, and revised.

  1. Please add the unit of y-axis of Fig. 7b-c. and decimal of Fig.7c should be only 1 decimal.

The word ‘intensity’ has been added to the unit of y-axis of Fig. 7b-c. However, changing the ppm value in 7c to a single decimal place does not align with the values in b and reduces accuracy. Therefore, I believe it is better to leave it as it is. Additionally, it is generally more appropriate to use two decimal places for ppm values in 1H NMR.

  1. Please check and correct the use of , at the end of title or journal name in all ref lists for consistency and follow the journal format. doi?
  2. Supplementary Materials:  ???

Response 13, 14: This feedback has been reviewed, and revised.

Round 3

Reviewer 2 Report

Comments and Suggestions for Authors

1. Please change the title to "Development of Dispersion Process to Improve Quality of Hyaluronic Acid Filler Crosslinked with 1,4-Butanediol Diglycidyl Ether" 

2. Please reveal the affiliations of all authors (1-4) or change intoappropriate form as journal format.

3. Line 10, "Hyaluronic" should be hyaluronic"???

4. Line 17, Therefore The viscoelasticity????

5.  Please consider for punctuation of these phases. "viscosity(cP)=24M: 43M, storage modulus(Pa)=306: 19 538, loss modulus(Pa)=33:61, Tack(N)=0.24:0.43) and factory scale (previous process: new process, 20 complex viscosity(cP)=19M: 26M, storage modulus(Pa)=229: 314, loss modulus(Pa)=71: 107, 21 Tack(N)=0.35: 0.43)." and so on other part of content and Tables.

6. Please rewrite into appropriate format small or capital letter or mixed "DAIHAN Scientific, Republic of Korea), and the crushed hydrogels 116 were sterilized using an autoclave (HS-3041VD, HANSHIN MEDICAL. CO.,LTD, Repub- 117 lic of Korea)"  " (OST-CMT1811, OSTAR TECH CO., Ltd,"   and so on.

7.  Figure 6(d)~(e)  ????? What is the meaning of  ~

8. Please remove the horizontal gride-line from Fig. 6. and Fig.7d.

Author Response

  1. Please change the title to "Development of Dispersion Process to Improve Quality of Hyaluronic Acid Filler Crosslinked with 1,4-Butanediol Diglycidyl Ether" 
  2. Please reveal the affiliations of all authors (1-4) or change intoappropriate form as journal format.
  3. Line 10, "Hyaluronic" should be hyaluronic"???
  4. Line 17, Therefore The viscoelasticity????
  5. Please consider for punctuation of these phases. "viscosity(cP)=24M: 43M, storage modulus(Pa)=306: 19 538, loss modulus(Pa)=33:61, Tack(N)=0.24:0.43) and factory scale (previous process: new process, 20 complex viscosity(cP)=19M: 26M, storage modulus(Pa)=229: 314, loss modulus(Pa)=71: 107, 21 Tack(N)=0.35: 0.43)." and so on other part of content and Tables.

Response 1-5: This feedback has been reviewed, and revised.

  1. Please rewrite into appropriate format small or capital letter or mixed "DAIHAN Scientific, Republic of Korea), and the crushed hydrogels 116 were sterilized using an autoclave (HS-3041VD, HANSHIN MEDICAL. CO.,LTD, Repub- 117 lic of Korea)"  " (OST-CMT1811, OSTAR TECH CO., Ltd,"   and so on.

Response 6: The phrases such as "CO. LTD.," have all been revised to "Co., Ltd.," as per the requested format. However, please note that company names have been left unchanged, as they are unique identifiers. Thank you for your understanding.

  1. Figure 6(d)~(e)  ????? What is the meaning of  ~
  2. Please remove the horizontal gride-line from Fig. 6. and Fig.7d.

Response 7, 8: This feedback has been reviewed, and revised.
